# Integrated miRNA Changes in Canine Testis and Epididymis According to Age and Presence of Cryptorchidism

**DOI:** 10.3390/ani13081390

**Published:** 2023-04-18

**Authors:** Eun Pyo Kim, Jae-Ho Shin, Wan Hee Kim, Geon A. Kim

**Affiliations:** 1Department of Theriogenology and Biotechnology, Research Institute for Veterinary Science, College of Veterinary Medicine, Seoul National University, Seoul 08826, Republic of Korea; 2Department of Biomedical Laboratory Science, College of Health Science, Eulji University, Seongnam 13135, Republic of Korea; 3Department of Veterinary Clinical Sciences, College of Veterinary Medicine and Research Institute for Veterinary Science, Seoul National University, Seoul 08826, Republic of Korea; 4Department of Biomedical Laboratory Science, College of Health Science, Eulji University, Uijeongbu 34824, Republic of Korea

**Keywords:** aging, miRNA, testis, cryptorchidism, testicular tumor

## Abstract

**Simple Summary:**

MicroRNAs have been actively studied for their diagnostic, prognostic, and therapeutic purposes for various diseases in animals. In this study, we analyzed the changes in miRNA expression under different conditions in the testis and epididymis of male dogs. We identified differentially expressed miRNAs in the testis and epididymis depending on age and the presence of cryptorchidism in male dogs, which suggests that they could help better understand aging and diseases of the reproductive system in male dogs.

**Abstract:**

In the present study, we aimed to investigate age-, cryptorchidism-, and testicular tumor-related changes in miRNAs in the testis and epididymis of dogs. Twelve healthy male dogs were divided into two groups: young (<1 year, n = 8) and old (>3 years, n = 4). Five dogs with unilateral cryptorchidism, one with a Sertoli cell tumor, and one with seminoma were referred to a veterinary hospital. After surgery, the testes and epididymis tails were collected. A high-throughput miRNA array analysis was performed to identify miRNAs affected by age, cryptorchidism, and testicular tumors. The expression of only cfa-miR-503 was downregulated in the epididymis of younger dogs, whereas the expression of 64 miRNAs was upregulated. Among them, the top five miRNAs were cfa-miR-26a, cfa-miR-200c, cfa-let-7c, cfa-let-7b, and cfa-let-7a. The expression of cfa-miR-148a and cfa-miR-497 was considerably lower in cryptorchid testis than in healthy dog testis. In the epididymis, the cfa-miR-1841 level was significantly decreased. We observed a significant difference in the expression of 26 cfa-miRNAs between testicular tumors and normal tissues. This study demonstrated that aging and cryptorchidism have a causal relationship with miRNA expression. The identified miRNAs may be candidate genes for male reproductive traits and could be applied in molecular breeding programs.

## 1. Introduction

Male reproductive function declines with age in both humans and dogs. It is well-known that testis volume, germ cell and Sertoli cell number, serum testosterone levels, and sperm viability decrease with age in humans [1]. Reproductive aging is associated with a reduction in the number of Leydig cells and seminiferous tubules, as well as oxidative stress [2,3]. In addition, changes in the expression of genes involved in the regulation of apoptosis and DNA repair have been observed in aging testis [4]. A negative correlation has been reported between age and normal ejaculated sperm characteristics in dogs [5]. A previous study has shown that epididymal sperm quality and fertility decrease with age in male dogs [6]. Furthermore, fresh and thawed semen from old male dogs display decreased sperm motility and mitochondrial function [7]. Therefore, age is one of the most important factors influencing infertility in male animals, including dogs. 

Cryptorchidism is the most common pathological condition in dogs, in which the testis fails to descend to the base of the scrotum, with an informed morbidity rate of 0.8–10% [8]. Hereditary defects caused by sex-limited autosomal recessive genes are considered an etiology of cryptorchidism in male dogs. Although cryptorchidism is often considered a mild malformation, it can seriously affect the health of dogs, as a very high-risk factor for sterility and testicular tumor. As cryptorchid testes generally have considerably higher temperatures than normally descended testes, unilateral cryptorchids reduce semen quality or cannot produce normal sperm [9]. Cryptorchidism is associated with testicular tumors in dogs [10,11]. The incidence of testicular tumors in undescended testes is approximately 13 times greater than that in normally descended testes [8]. Although testicular tumors are typically malignant, the excess production of endogenous estrogen by the tumor cells can lead to a condition known as feminization syndrome and bone marrow suppression, which can be fatal [12]. 

MicroRNAs (miRNAs) are a group of short non-coding RNA molecules and are approximately 20–24 nucleotides long; they function by epigenetically downregulating gene expression. miRNAs play a crucial role in controlling both mRNA stability and protein synthesis, thereby influencing the processes of spermatogenesis [13] and sperm maturation during sperm passage through the epididymis [14]. Although the differentially expressed miRNAs in the testicular tissue in cryptorchidism have been investigated in humans and various animal species, including horses, rats, and mice [15], direct correlation studies between miRNAs and cryptorchidism in dogs are limited. miRNAs can be utilized as biomarkers to elucidate changes in tissues affected by several disorders. In our previous study [16], we identified several miRNAs associated with uteropathies and age markers in female dogs. To the best of our knowledge, there are no reports of alterations in miRNA expression within the testes and epididymis of canines of different ages or with cryptorchidism and testicular tumors. The purpose of this study was to (i) assess the relationship between aging and disease status and the expression of miRNAs in male dog testis and epididymis and (ii) determine the potential of miRNAs as biomarkers and therapeutic targets for infertility and various diseases in male dogs.

## 2. Materials and Methods

### 2.1. Collection of Tissue Sample 

Male reproductive organ tissues were collected from two local animal hospitals and the veterinary teaching hospital of Seoul National University, Korea, at the request of the dog owners when general neutralization surgery or castration was required according to clinical findings, such as cryptorchidism, Sertoli cell tumor, and seminoma, with owners’ informed consent. Testicular tumor diagnosis was based on histological examination (IDEXX, Seoul, Republic of Korea) of the surgically removed tissues. Furthermore, testicular tumor tissue collection was approved only by the Seoul National University Institutional Animal Care and Use Committee (approval number: SNU-200217-3-2). The dogs with testicular tumors were not in the cryptorchidism state. The testes were obtained from 19 male dogs and epididymis cauda tissues (tail part of the epididymis) were collected from 17 male dogs, excluding two dogs with testicular tumors, during castration for the prevention and treatment of disease at the request of the owners, with informed consent obtained from the owners. In dogs, the final diagnosis of cryptorchidism can only be made after 6 months of age [17]. All cryptorchid testes were palpable in the right inguinal area and the left testis was present within the scrotum. Before anesthesia for castration, all dogs underwent appropriate clinical evaluations such as blood chemistry tests for total protein, glucose, blood urea nitrogen (BUN), creatinine, alkaline phosphatase (ALKP), and alanine aminotransferase (ALT) to identify any potential pathological conditions and determine their suitability for systemic anesthesia. The serum chemistry results for the dogs used in this clinical examination are shown in Table 1. This study involved eight healthy male puppies under the age of 1 year, four healthy male dogs over the age of 3 years, five dogs with cryptorchidism under the age of 1 year, one dog with Sertoli cell tumor, and one dog with seminoma over the age of 10 years. The average results and reference ranges of each parameter of the serum chemistry are presented in Table 1. Before the tissues dried or were damaged, each testis and tail of the epididymis were promptly separated after surgery and collected. As the testis and epididymis are surrounded by the tunica vaginalis, removal of the tunica vaginalis was performed after scrotal incision. The immature spermatozoa within the testes gain the ability to move and become capacitated as they pass through the epididymis. Therefore, we collected the testes and tail parts of the epididymis cauda to compare miRNA expression in all tested dogs. To collect the epididymis cauda tissue, the surrounding tissues such as the ligament of the epididymis were excised and separated from the testis. Some of the cryptorchid and descended tissues of the testis and epididymis were stored in neutral buffered formalin for histological examination. In addition, the remaining testis and epididymal cauda tissues from each male dog were collected in RNAlater (Thermo Fisher, Waltham, MA, USA) and stored at −80 °C for miRNA array analysis.

### 2.2. Hematoxylina and Eosin Staining 

Both cryptorchid and non-cryptorchid sides of the testes and epididymis of dogs with unilateral cryptorchidism were collected for hematoxylin and eosin (H&E) staining and analysis. The testis and epididymis cauda were fixed in neutral buffered formalin, dehydrated with a gradient series of alcohol from 60% to 90%, and embedded in paraffin. According to the standard protocol, H&E-stained 4-µm sections (Leica Microsystems GmbH, Wetzlar, Germany) placed on silane-coated slides were observed under a microscope (BX53; Olympus, Tokyo, Japan).

### 2.3. RNA Isolation and Quality Check, and cDNA Synthesis 

The total RNA was extracted following the protocol provided by the manufacturer. To extract the total RNA from each tissue sample, weighing a minimum of 50 mg, an easy-spin™ Total RNA Extraction Kit (Intron Biotechnology, Seoul, Republic of Korea) was used to homogenize the samples with 1 mL of RNA lysis solution. To perform gene microarray hybridization, the Agilent RNA 6000 nano kit and 2100 Bioanalyzer (G2939BA; Agilent, Santa Clara, CA, USA) were used to assess both the quantity and quality of the RNA. Only samples that fulfilled the following criteria were selected for microRNA analysis: A260/A280 and A260/A280 > 1.0, concentration > 50 ng/µL, volume > 10 µL, total amount > 0.7 µg, rRNA ratio > 1.0, and RIN > 7.0 with visible small RNA peaks. Before cDNA synthesis, the NanoDrop 2000 spectrophotometer (Thermo Fisher Scientific Inc., Wilmington, MA, USA) was used to assess the quantity and quality of the RNA. The Maxime RT-PCR premix kit (Intron Biotechnology) was used to reverse transcribe the RNA samples (500 ng) to cDNA, with a reaction mixture of a total volume of 20 µL.

### 2.4. MicoRNA Hybridization, Scanning, and Data Processing

All procedures were performed in accordance with the guidelines in the Affymetrix Expression Analysis Technical Manual (Affymetrix Inc., Santa Clara, CA, USA). Total RNAs were labeled using the FlashTag Biotin HSR RNA Labeling Kit (Thermo Fisher). The GeneChip^®^ Affymetrix miRNA microarray (Affymetrix Inc.) was used to hybridize the biotin-labeled samples. The Affymetrix GeneChip^®^ scanner was used to examine all arrays, and raw data analysis was carried out using the Affymetrix GeneChip^®^ Command Console^®^ Software (AGCC) (Affymetrix Inc., Santa Clara, California, USA). The measured intensity of each array probe was obtained from CEL files, which contained the raw data images developed by the scanner. A *p*-value of <0.05 and fold change of >2 served as cut-off criteria to restrict a broad range of differentially expressed microRNAs. 

### 2.5. Statistical Analysis

The clinical data of the dogs are presented as mean ± standard deviation. An unpaired *t*-test was used to compare the groups. All statistical analyses were conducted with SPSS software (version 25.0; SPSS Inc., Chicago, IL, USA). The level of statistical significance was defined as *p* < 0.05.

## 3. Results

### 3.1. H&E Staining of Cryptorchid Testis and Epididymis

The cryptorchid side testis showed severe diffuse tubular atrophy of the seminiferous tubules and only Sertoli cells with germ cell depletion (Figure 1A). However, the other (non-cryptorchid) side testis showed tubular atrophy, which was less severe than that of the cryptorchid-side testis (Figure 1B). Degeneration of germ cells and depletion of elongating spermatids were observed. In the cryptorchid-side epididymis, no spermatozoa showed epithelial tubule degeneration (Figure 1C). Cell debris and germ cell exfoliates were observed; however, no spermatozoa were observed in the lumen of the non-cryptorchid side epididymis (Figure 1D).

### 3.2. MiRNA Expression in the Testis and Epididymis Cauda in the Dogs according to Age

The testis and epididymis cauda miRNA volume plots for the comparison of immature male dogs younger than 1 year of age versus mature dogs older than 3 years of age are presented in Figure 2. According to the fold change cut-off of >2 (upregulation or downregulation) and *p* value of <0.05, there were no significantly differentially expressed miRNAs in the testes between immature and mature dogs. In the epididymis, the top five miRNAs with noticeable differential expression were confirmed by considering the fold change, *p*-value, and volume value. Sixty-five significant miRNAs that were differentially expressed were identified. Among them, the top five meaningful miRNAs, considering fold change and volume values, were cfa-miR-26a, cfa-miR-200c, cfa-let-7c, cfa-let-7b, and cfa-let-7a. The only significant miRNA that was less expressed in mature dogs compared with that in immature dogs was miR-503.

### 3.3. MiRNA Expression of Testis and Epididymis Cauda in Cryptorchid Dogs

As shown in Figure 3A, two significantly downregulated miRNA genes, cfa-miR-148a and cfa-miR-497, were found in the testes of cryptorchid dogs compared with age-matched normal dogs. As shown in Figure 3B, cfa-miR-1841 was significantly downregulated in the epididymis of cryptorchid dogs.

### 3.4. MiRNA Expression in the Testis of Dogs with Sertoli Cell Tumor and Seminoma

One hundred and seven miRNAs, with significant differences in expression between Sertoli cell tumors and normal testes, were identified. Sixteen miRNAs were downregulated, and most of the remaining ninety-one miRNAs were upregulated in one case of Sertoli cell tumor compared with those in the normal testis. The top five miRNAs with a fold change cut-off of >2 (upregulation or downregulation) between Sertoli cell tumor and normal testis were cfa-miR-27b, cfa-miR-20a, cfa-miR-93, cfa-miR-502, and cfa-miR-500 (Appendix A). Cfa-miR-27b, cfa-miR-20a, and cfa-miR-93 were highly expressed in Sertoli cell tumors, and cfa-miR-502 and cfa-miR-500 were less expressed. In seminoma, 86 miRNAs, different from those in normal testes, were identified. It was confirmed that 21 miRNAs were downregulated and 65 were highly expressed. The top five miRNAs with a fold change cut-off of >2 (upregulation or downregulation) were cfa-miR-378, cfa-miR-29a, cfa-miR-27b, cfa-miR-16, and cfa-miR-106a (Appendix A). The overlapping and non-overlapping differentially expressed miRNAs between the Sertoli cell tumor and seminoma groups are presented in Appendix A as Venn diagrams. The miRNAs indicated in red were differentially upregulated and those in blue were downregulated. The miRNAs indicated in green were upregulated in one comparison, which showed contradictory results for each comparator group, but downregulated in the other comparison. While there was an overlap between 55 significant miRNAs in “Sertoli cell tumor vs. normal testis” and “Seminoma vs. normal testis,” the expression of 11 miRNAs indicated in green were interestingly discordant in each comparison, indicating that a significant portion of the “tumor type” could affect the miRNA expression pattern. Twenty-nine miRNAs were commonly identified in the comparison between Sertoli cell tumor and normal testis and between seminoma and normal testis. Next, both Sertoli cell tumors and seminomas, regarded as testicular tumors, were grouped and compared to age-matched normal testes. Significant changes in the expression of 28 miRNAs were observed in testicular tumors compared with that in the normal testes. All 28 miRNAs (100%) overlapped, with the expression of the miRNAs frequently changing compared with that in the comparison of Sertoli cell tumors versus normal testes. Two miRNAs, miR-216b and miR-449, were commonly downregulated in the comparisons of testicular tumors versus normal testes and Sertoli cell tumors versus normal testes. The overlapping parts of the three circles represent 26 miRNAs—23 upregulated and 3 downregulated. 

## 4. Discussion

The results of this study provide conclusive evidence regarding the effect of age and cryptorchidism on miRNA expression in the male reproductive system. The miRNA expression in the epididymis observed in this study was affected by age, with increased expression of cfa-let-7 members, cfa-miR-26a, and cfa-miR-200c in dogs older than 3 years of age. We focused on miRNA expression in dogs with cryptorchidism and testicular tumors as (1) cryptorchidism is the most common hereditary disorder in male dogs and (2) it may be a risk factor for infertility and testicular cancer. Our results showed that in cryptorchid dogs, the expression of cfa-miR-148a and cfa-miR-497 was downregulated in the testes and that of cfa-miR-1841 was downregulated in the epididymis compared with those in age-matched dogs. In addition, testicular tumors showed different miRNA expression levels compared with those in age-matched dogs. 

In addition to advanced maternal age, paternal age tends to be significantly associated with a decline in undesirable embryonic development and poor pregnancy outcomes because semen volume, sperm motility, and normal morphology can significantly decline with age in humans [18]. Similarly, in various animals, including dogs, ferrets, and cats, age negatively correlates with male reproductive capacity [5,6,19,20]. To the best of our knowledge, there is no report on the miRNA expression patterns in the reproductive organs of dogs across different age groups. In this study, the dogs were categorized into two distinct age groups (<1 year old and >3 years old) for the purpose of evaluating miRNA expression in the testes and cauda epididymis. Interestingly, the miRNA levels were not significant in the testis but were significant in the epididymis, with higher expression of cfa-miR26a, miR-200c, and let 7 family members (let 7a, let7b, and let 7c). These results suggest that the testis and epididymis are different in terms of miRNA expression and processing. 

The epididymis serves as the location for the post-maturation of testicular sperm, resulting in the acquisition of sperm motility and the capacity to recognize and fertilize oocytes. In this study, epididymal tissue was harvested from the cauda epididymis, where mature sperm are stored until ejaculation. To minimize the influence of prostatic fluid and immature sperm, the cauda epididymis may be a more appropriate tissue to examine the effects of age and cryptorchidism on miRNA expression in dogs. Furthermore, as the incidence of canine prostatic disorders markedly increases with age, the miRNA expression results may have been affected by a mixture of prostatic fluid. 

miR-26a is a functional miRNA that regulates sperm metabolism and apoptosis. It has been reported that miR-26a can have a major effect on the quality of semen in Holstein bulls as it plays a negative regulatory role in the expression of phosphoenolpyruvate carboxykinase-1 (PCK1) [21]. Additionally, it is suggested that miR-26a is involved in the regulation of bull sperm motility [22]. In boars, miR-26a is associated with decreased sperm motility [23]. A previous study demonstrated that there is a considerable increase in the expression of miR-26a in highly motile frozen-thawed sperm compared with that in low-motile frozen-thawed sperm [24]. Furthermore, the sperm transcript level of miR-26a-5p is lower in men with unexplained infertility than in fertile control, and high phosphatase and tensin homolog (PTEN) expression is associated with ejaculated spermatozoa [25]. PTEN signaling is a major negative regulator of PI3K signaling and is involved in the maintenance of spermatogonial stem cells in mice [26]. Interestingly, our study showed that cfa-miR-26a was highly expressed in the epididymis of mature dogs compared with that in immature dogs. These findings offer compelling evidence supporting the hypothesis that miR-26a may play a role in the regulation of epididymal aging and, consequently, have an effect on sperm metabolism or motility. 

Let-7 (lethal-7) is among the earliest identified miRNAs. As the expression of let-7 family members gradually increases during development, it is not surprising that high levels were observed in the cauda epididymis of mature dogs compared with those of younger dogs. However, in a previous study, female germ cells did not show changes in let-7 miRNA expression, but male germ cells showed increased expression during development [27]. Boars with low sperm motility and a high percentage of abnormal sperm showed higher levels of let-7a, let-7d, and let-7e miRNAs in their spermatozoa [28], indicating that the let-7 family members may be markers for infertility. Interestingly, although the expression of miR-26a and let-7 family members was increased in the ovaries of dogs with uteropathies in our previous study, significant differences in the expression of these miRNAs were observed in the epididymis of mature dogs in this study. Therefore, miR-26a and the let-7 family members may be related to female reproductive organ disease in dogs as well as the aging of male reproductive organs. In Yorkshire boars, endogenous miR-26a and let-7 have anti-apoptotic and pro-survival functions in sperm cells by targeting *PTEN* and *PMAIP1* [23]. The most abundant miRNAs in the epididymis of bovine species are let-7 family members and miR-200a/b tumor suppressors [29]. miRNA-mediated inactivation of cellular oncogene products could play a role in maintaining the stability of the epididymis, which is an organ with a unique ability to evade tumorigenicity. Consistent with these findings, the let-7 family members and miR-200a/b tumor suppressors, whose expression was confirmed in the epididymis of younger dogs, were not identified in the testicular tumors in the present study. 

The miR-200 family, which includes miR-141, miR-200a, miR-200b, miR-200c, and miR-429, is the most prevalent family in the miRNA system, and all members are highly conserved. It has been reported that the miR-200 family members regulate epithelial–mesenchymal transition, which is an important step for breast cancer infiltration and metastasis [30]. Furthermore, miR-200 has a functional role in the regulation of cell invasion and migration by targeting PTEN [31]. The expression of miR-200a, miR-200c, and miR-141 in both male and female mouse germ cells is downregulated gradually during development [27]. In a human study, miR-200a-3p and miR-200c-3p were identified as potential biomarkers for male subfertility [32]. As miR-200b and miR-200c accumulate in spermatozoa during passage through the epididymis [33,34], the present study results indicate that sperm maturation in the epididymis cauda may be differentially regulated according to dog age. 

Cryptorchidism is a congenital defect commonly found in dogs, where one or both testicles fail to descend normally into the scrotum. The retained testicle(s) may remain in the abdomen or become lodged in the inguinal canal, causing potential health issues such as infertility, testicular tumors, and torsion. In recent years, several studies have focused on miRNA expression in the testes of various animals with cryptorchidism, including horses [35], rats [36], and mice [37]. miR-148a was identified as differentially expressed in the seminal plasma extracellular microvesicles of men with oligoasthenozoospermia subfertility compared with that in men with normozoospermic fertility, demonstrating that it may be a marker for male infertility [38]. Similarly, the present study suggests that cryptorchid-side testes and epididymis do not have spermatozoa and cryptorchid-side testes display differential expression of miR-148a. In agreement with the fact that miR-497-5p has been identified only in the testicular tissues of spermatozoa and seminal plasma in humans [39], cfa-miR-497 was found to be expressed at low levels in the testes of cryptorchid dogs. 

In testicular tumors, miR27b was highly expressed in both Sertoli cell tumors and seminomas; it has been found to be differentially expressed in the mature spermatozoa of infertile men. The high expression of cysteine-rich secretory protein 2 (CRISP2), which is predominantly expressed in the testis, is correlated with the expression of miR-27 in humans [40]. miR-27b expression is downregulated during sheep fetal testis development from D42 to 75 and plays an important role in regulating cellular differentiation [41].

A major limitation of this study is that the miRNA expression was analyzed in a limited number of dogs, especially in one case each of Sertoli cell tumor and seminoma. The present analysis of miRNA expression in testicular tumor tissues lays a foundation for more extensive and larger-scale studies. The results of this study should be interpreted with caution. Furthermore, age and breed predisposition may have affected the miRNA expression results of this study. In order to reduce this expected bias, we classified the dogs into two age categories, those under 1 year old and those over 3 years old; evaluated tissue samples obtained only from small dog breeds; and analyzed age-matched dogs as a control group for the comparison of cryptorchidism testes and testicular tumors. 

## 5. Conclusions

In conclusion, miRNA expression in the male reproductive tissue of dogs with cryptorchidism and testicular tumors was comprehensively analyzed and the effect of age on miRNA expression in male reproductive organs was examined. The present study demonstrated that epididymal cfa-miR-26a, cfa-miR-200c, cfa-let-7c, cfa-let-7b, and cfa-let-7a expression regulated via PTEN may be involved in aging in dogs. Furthermore, cfa-miR-148a and -497 expression were consistently lower in the testes and cfa-miR-1841 expression was lower in the epididymis of dogs with cryptorchidism, suggesting that these miRNAs may be useful biomarkers for cryptorchidism and male infertility. This study provides insights into the development and causes of cryptorchidism and testicular tumors. We hope that our study will help develop new diagnostic methods and preventive medication for spontaneous cryptorchidism and testicular tumors and improve male fertility with aging in dogs. 

## Figures and Tables

**Figure 1 animals-13-01390-f001:**
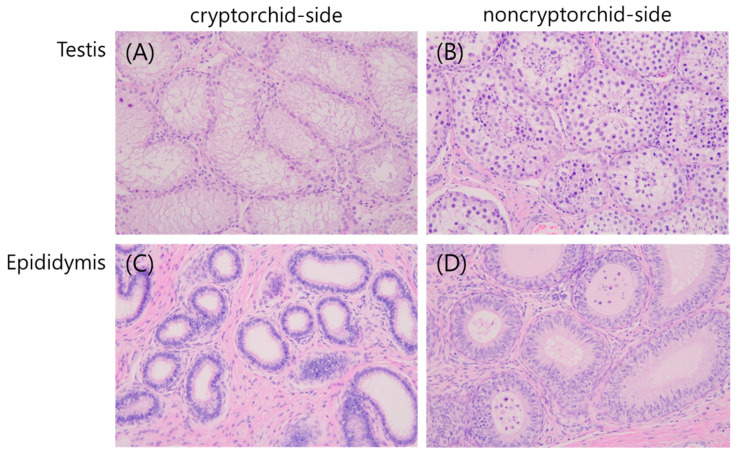
Images of hematoxylin and eosin-stained sections of the testis (**A**,**B**) and epididymis (**C**,**D**) in dogs with unilateral cryptorchid. The reproductive organs from the cryptorchid (**A**,**C**) and non-cryptorchid (**C**,**D**) sides were compared.

**Figure 2 animals-13-01390-f002:**
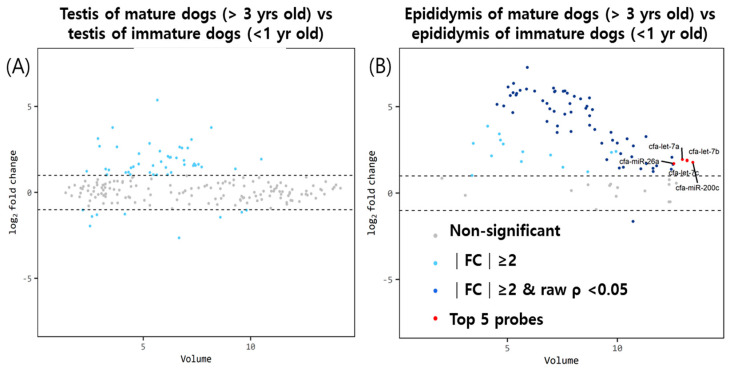
Volume plots of miRNAs in the (**A**) testis and (**B**) epididymis of mature 3−year−old dogs (above) compared with those of immature 1-year-old dogs (below). The volcano plot illustrates the fold changes and volumes between the healthy immature and mature dogs. The horizontal line indicates a 2−fold difference. The red dots indicate the significance of the top five miRNAs for comparison. The blue and sky-colored dots represent significant differences above a 2−fold change without a *p*-value of <0.05. Cfa-miR = *Canis lupus familiaris* microRNA. Gray dots signify no meaningful correlation between volume and fold change.

**Figure 3 animals-13-01390-f003:**
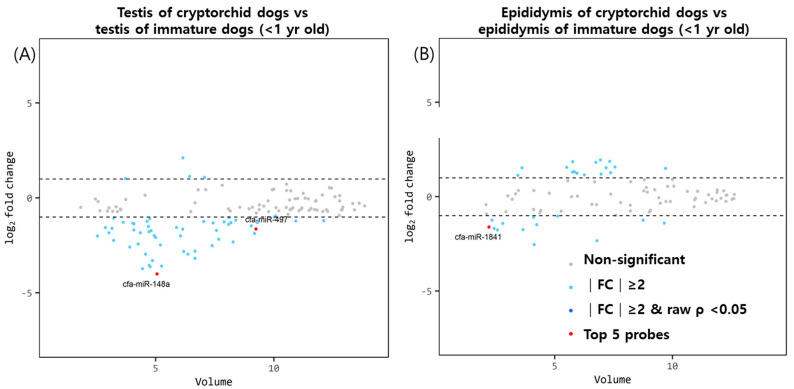
Volume plots of miRNAs in the (**A**) testis and (**B**) epididymis of cryptorchids dogs compared with dogs younger than 1 year of age. The volcano plot illustrates the fold changes and volumes between the groups. The horizontal line indicates a 2−fold change difference. The red dots indicate the top five significant miRNAs for comparison. The blue and sky−colored dots represent significant differences above a 2-fold change without a *p*-value of <0.05. Cfa-miR = *Canis lupus familiaris* microRNA. Gray dots signify no meaningful correlation between volume and fold change.

**Table 1 animals-13-01390-t001:** Clinical examination of dogs included in this study.

Variable (Mean ± SEM)	Healthy Dogs, below 1 Year of Age (n = 8)	Healthy Dogs, above 3 Years of Age (n = 4)	Dogs with Cryptorchidism (n = 5)	Dogs with Sertoli Cell Tumor (n = 1)	Dogs with Seminoma (n = 1)	Reference Value
Age, months	5.9 ± 0.7	80.0 ± 13.4	7.8 ± 1.0	120	144	
Breed	papillon (1), poodle (2), Maltese (1), bichon fries (1), Pomeranian (2), shih tzu (1)	poodle (1), mixed (2), Maltese (1)	Maltipoo (1), poodle (3), Maltese (1)	Chihuahua	Pomeranian	
Total protein (g/dL)	5.6 ± 0.2	6.9 ± 0.4	5.8 ± 0.1	6.27	6.8	5–7.2
Glucose (mg/dL)	115.6 ± 3.4	120.5 ± 9.4	114.2 ± 5.2	97	154	75–128
BUN (mg/dL)	22.0 ± 2.2	15.8 ± 1.9	20.5 ± 4.3	17.5	18	9.2–29.2
Creatinine (mg/dL)	0.5 ± 0.1	0.7 ± 0.1	0.5 ± 0.1	0.77	0.5	0.4–1.4
ALKP (U/L)	494.7 ± 137.4	108.0 ± 25.0	365.4 ± 47.6	46	83	47–254
ALT (U/L)	45.8 ± 3.1	84.5 ± 32.7	66.0 ± 7.8	50	171	17–78

## Data Availability

Not applicable.

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
