# Peer review of "Integrated miRNA Changes in Canine Testis and Epididymis According to Age and Presence of Cryptorchidism"

_animals, 2023, doi:10.3390/ani13081390_

Round 1

Reviewer 1 Report

The submission "Integrated miRNA in canine testis and epididymis according to age, cryptorchidism, and testicular tumor"  brings molecular information of main mecanisms that corroborate to male infertility. The introduction is well constructed according to the literature. The material and methods are well applied and described. Moreover, it is capable to be reproduced. The results are clear and discussion compare important data among a diversity of studies. In conclusion, the manuscript is acceptable for publication.

The submission "Integrated miRNA in Canine Testis and Epididymis according to Age, Cryptorchidism, and Testicular Tumor" addressed the importance of tumor of testicular. 
The topic is original.

It is an innovative idea in the field.

Everything is ok regarding the methodology.  
The conclusions are consistent.

The references are appropriate.

Author Response

Thank you for giving me a good evaluation of my manuscript.

I will try to refine and supplement it better so that it can be published as a good paper.

Thank you very much.

Reviewer 2 Report

The present manuscript aims to analyze overall miRNA expression in testicular and epididymal tissue according to age group, cryptorchidism and testicular tumor in dogs. The main goal was to contribute to the understanding of the physiology of ageing sperm-related changes, as well testicular disorders. For that purpose, dogs were orchiectomized and testis and epididymides were analyzed through for a general panel of miRNA expression. This is an important research basic area and the questions asked are relevant for small animal theriogenology and reproductive physiology.

This manuscript is interesting, but I am not sure if it is meant to be a preliminary study or an original research article. If it is the latter, the authors will need to provide more supporting evidence to previous publications in this area of research. The unique analysis of testicular and epididymides miRNA expression is not enough to attest any relationship between age and testis disorders, as the protein level was not identified. More importantly, the identification of miRNA expression of a more robust number of subjects with testicular tumor is lacking. Although testicular disorders are common concerns in canine practice, and deserve profound scientific investigations, this manuscript is based on an experimental design with important concerns and methodological issues that impair data reliability. Therefore, all data regarding testicular tumor should be suppressed in this manuscript, because results based on only one subject are not representative of the disorder itself. Unfortunately, we cannot rely on the results generated by only 1 dog and these data must be disregarded. Some areas of concern and methodological issues should be addressed. Some of the major comments and criticisms include:

Title: delete “testicular tumor”.

Simple summary:

All these information has to be useful for non-researchers. Therefore, the terminology is not adequate.  

Summary:

Lines 18: only epididymis? and in the testis as well?

Lines 22-28: the simply quotation of the miRNA is not at all understandable and useful for abstract readers.

Line 28: the older dogs cannot be considered senescent animals. Please, reconsider

Lines 28-29: Is the association between miRNA and ageing / testicular disorders suggested to be causative or consequence?

Introduction:

More information should be provided regarding the mechanisms involved on reproductive ageing and etiology of cryptorchidism.

Line 36-37: in which species?

The manuscript lacks an objective. Authors should clearly state the aims of the research.

Material and methods:

Please, describe in detail the subjects you used to compose the groups. What was, in fact, the experimental groups?

Lines 77-78: please describe more accurately, what each dog was diagnosed for? How many cryptorchidic dogs had also testicular tumors?

Lines 80-82: were all dogs unilateral cryptorchidic? There was no intra-abdominal testis?

Lines 93-97: very confuse description of the experimental design. How many testicules and epididymides were histologically examined? How many samples were subjected to miRNA analysis?

Lines 131-132: The clinical data are not considered variables data, but only classificatory data and/or inclusion criteria. 

Results:

If you want to include the clinical data, you should provide as a description of your study population, confirming the inclusion criteria. Clinical data is a classificatory variable and not a response variable and, thus, these results were only used to allocate animals within certain groups. This should not be placed in the Result section.

Lines 138-140: this should be stated in the MM section. To which group all these dogs were assigned to? Aside for being under 1 year old, were all dogs post-pubertal, i.e., between 6/7 months and 12 months?

It is not acceptable to have only one subject in each testicular tumor type.

Table 1: Despite the fact that clinical information is duplicated in the text and in Table 1, all table content should be stated in the MM section.

Lines 185: try to be consistent with the way you named the experimental groups (young X older or mature X immature?)

Lines 192-193: the expected comparison to be made is between the retained testis and its contralateral scrotum testis. That means that each dog is its own experimental control.

Lines 202-248: unfortunately, we cannot rely on the results generated by only one subject. Please, these data must be disregarded.

Discussion

Unfortunately, discussion is poor, because the experimental design does not allow any further inferences. Authors can give a special attention to the molecular pathways involved on each miRNA. The protein or group of proteins translated by each miRNA should be taken into account in the discussion.

Lines 261-265: This only regards to senescence, i.e., dogs that are beyond the reproductive age (6-7 years depending on body weight), which is not the case for this research.

Lines 282-284: in which tissue? Testicular or epididymal miR-26a?

Conclusion

Lines 348-349: please, be more specific on the molecular pathways involved with all these miRNAs.

Author Response

First of all, thank you very much for your good points and advice. I tried to do an original study because research on the expression of miRNA according to age and disease in testis and epidididymis of male dogs has not yet been actively conducted. Unlike groups in according to age and cryptorchidism, we know that populations of testicular tumors group are significantly insufficient and so their limitations are specified in the text. In our country, castration in young individuals is becoming more common, so meeting testicular tumor patients in actual clinical trials is gradually decreasing. But even a very small group of experiments might find meaningful clues, so we used them to analyze them. In future research projects, we plan to gather more patients with testicular tumors to create more significant data and write additional papers. As you pointed out, the part about testicular tumors will be excluded from the title and the contents will be reflected for reference in future studies.

Title: delete “testicular tumor”.

Answer: I deleted it as you pointed out.

Simple summary:

All this information has to be useful for non-researchers. Therefore, the terminology is not adequate.  

Answer: I've refined the terms and sentences as follows.

MicroRNA has been actively studied for diagnostic, prognostic, and therapeutic purposes of various diseases in animals. In this study, we analyzed miRNA changes according to various conditions in testis and epididymis of male dogs. Differentially expressed miRNAs in testis and epididymis were discovered depending on age and cryptorchidism in male dogs suggesting that they could be important potentials for understanding aging and disease in male dog reproductive system.

Summary:

Lines 18: only epididymis? and in the testis as well?

Answer: The word ‘testis’ is missing. I added it again and filled it out.

Lines 22-28: the simply quotation of the miRNA is not at all understandable and useful for abstract readers.

Answer: Since the important point of this study was the various expressions of the described miRNA, we had no choice but to list it. If you suggest the delete the simple quotation of the miRNA, we will remove them in our manuscript.

Line 28: the older dogs cannot be considered senescent animals. Please, reconsider

Answer: The word "senescence" simply meant aging. I changed it to "aging" because it could be confusing.

Lines 28-29: Is the association between miRNA and ageing / testicular disorders suggested to be causative or consequence?

Answer: Based on the specific mechanisms of miRNA, it is considered to be the cause of them.

Introduction:

More information should be provided regarding the mechanisms involved on reproductive ageing and etiology of cryptorchidism.

Answer: I have added the corresponding content and also added a reference.

→ Reproductive aging is associated with a reduction in the number of Leydig cells and seminiferous tubules, as well as oxidative stress. In addition, changes in the expression of genes involved in the regulation of apoptosis and DNA repair have been observed in the aging testis.

→ It is considered that hereditary defects caused by sex-limited autosomal recessive genes is etiology cryptorchidism of male dog.

Line 36-37: in which species?

Answer: This is from the article, "Effect of body weight, age and bleeding history on canine sperm quality parameters measured by the Hamilton-Thorne annalser." I added "in dogs" to the sentence in line 43-44.

The manuscript lacks an objective. Authors should clearly state the aims of the research.

Answer: The last sentence of the introduction was changed so that the purpose of the study could be clearly revealed in line 73~75.

The purpose of this study is to find the relationship between aging and disease status and expression of miRNA in male dog reproductive tissues and to use it as biomarkers and therapeutic targeting for fertility and disease prevention of male dog.

Material and methods:

Please, describe in detail the subjects you used to compose the groups. What was, in fact, the experimental groups?

Answer: The information about the experimental subjects and the groups is in the results section, and as you pointed out, the contents were transferred to the materials and methods section.

Lines 77-78: please describe more accurately, what each dog was diagnosed for? How many cryptorchidic dogs had also testicular tumors?

Answer: Each dog was diagnosed with cryptorchidism for preventive purposes and commissioned surgery. Dogs with testicular tumors were diagnosed for their own removal and performed surgery at the request of their owners. I inserted the content into the sentence.

The cryptorchidism dogs used in this study did not have testicular tumors, and the patients with testicular tumors were not in cryptorchidism state.

All testes were obtained from all 19 male dogs, and epididymis cauda tissues (tail part of epididymis) were collected from 17 male dogs during castration at the request of owners with informed consent from the owner.

  → All testes were obtained from all 19 male dogs, and epididymis cauda tissues (tail part of epididymis) were collected from 17 male dogs except for two testicular tumor dogs during castration for the prevention and elimination of disease at the request of owners with informed consent from the owner.

Testicular tumor diagnosis was based on histological examination (IDEXX, Republic of Korea). Also, testicular tumor tissue collection was approved by the Seoul National University Institutional Animal Care and Use Committee (approval number: SNU-200217-3-2).

  → Testicular tumor diagnosis was based on histological examination (IDEXX, Republic of Korea). Also, testicular tumor tissue collection was approved by the Seoul National University Institutional Animal Care and Use Committee (approval number: SNU-200217-3-2). The dogs with testicular tumors were not in cryptorchidism state.

Lines 80-82: were all dogs unilateral cryptorchidic? There was no intra-abdominal testis?

Answer: All cryptorchidism dogs were unilateral, and only subcutaneous cryptorchidism dogs were used.

Lines 93-97: very confuse description of the experimental design. How many testicules and epididymides were histologically examined? How many samples were subjected to miRNA analysis?

Answer: The number of samples used for miRNA analysis is as follows. All epididymis and tests of described dogs were examined for miRNA analysis. Overall 19 testes (healthy dogs below 1 year n=8; healthy dogs above 3 year n=4; cryptorchid side of cryptorchidism dogs n=5; Sertoli cell tumor n=1; Seminoma n=1) were examined for miRNA analysis. In case of epididymis, only 17 epididymis were examined for miRNA analysis. For identify the testicular tumor status, only two suspected testes were examined for histological analysis. In case of cryptorchidism, testis and epididymis of 5 dogs were examined for histological comparison between cryptorchid side and non-cryptorchid side. So cryptorchid side testis of cryptorchid dogs were dissected longitudinally into 2 pieces and half were fixed in neutral buffered formalin and another half were frozen for miRNA analysis.  Representative figures of H&E stain were inserted in our manuscript.

Lines 131-132: The clinical data are not considered variables data, but only classificatory data and/or inclusion criteria. 

Answer: If there were no other metabolic diseases or abnormal values through serum chemistry, they were included as normal subjects. Abandoned dogs were excluded because their age and breed was not clearly known, and only cases where the owner could clearly identify the age and breed of each dogs were included in this study.

Results:

If you want to include the clinical data, you should provide as a description of your study population, confirming the inclusion criteria. Clinical data is a classificatory variable and not a response variable and, thus, these results were only used to allocate animals within certain groups. This should not be placed in the Result section.

 Answer: As you mentioned above, I moved the contents to the M&M part. As we mentioned above, only dogs with clear information such as breed and age were used and abnormalities were confirmed through clinical examination.

Lines 138-140: this should be stated in the MM section. To which group all these dogs were assigned to? Aside for being under 1 year old, were all dogs post-pubertal, i.e., between 6/7 months and 12 months?

Answer: As you pointed out above, I moved the clinical data part to the M&M part and described it. Also, a group of young dogs under the age of 1 recruited dogs between 5 and 12 months old regardless of whether they were puberty or not.

It is not acceptable to have only one subject in each testicular tumor type.

Answer: As you mentioned, we understand that the results of one case of each testicular tumor analysis cannot be generalized. However, this is considered to be basic data for conducting additional studies using this as a pilot study and analyzing the associations with other common tumors in dogs. As you pointed out, we remove “testicular tumor” in the title.

Table 1: Despite the fact that clinical information is duplicated in the text and in Table 1, all table content should be stated in the MM section.

Answer: As you mentioned above, both the contents and table 1 were moved to the M&M part.

Lines 185: try to be consistent with the way you named the experimental groups (young X older or mature X immature?)

Answer: As you pointed out, we revised it to "mature and immature" for consistency in line 197.

Lines 192-193: the expected comparison to be made is between the retained testis and its contralateral scrotum testis. That means that each dog is its own experimental control.

Answer: As you pointed out, it is correct to compare the normal testicle and retained testicle of cryptorchid dogs. However, in this study, it was histologically confirmed that the normal testis and epididymis of dogs with unilateral cryptorchidism were normal and the testis and epididymis of a normal dogs of same age were used as a control group. This study is a pilot study, and in the next, we would like to compare the miRNAs in the bloods and testis of dogs with cryptorchid (unilateral or bilateral) and miRNAs from normal dogs and this study is considered to be basis for follow-up studies. 

Lines 202-248: unfortunately, we cannot rely on the results generated by only one subject. Please, these data must be disregarded.

Answer: Yes, as you pointed out, we disregard this data. We will move to supplementary data.

Discussion

Unfortunately, discussion is poor, because the experimental design does not allow any further inferences. Authors can give a special attention to the molecular pathways involved on each miRNA. The protein or group of proteins translated by each miRNA should be taken into account in the discussion.

Answer: As you pointed out, information on miRNA-related pathways has been generally added. It was confirmed that miRNA26a and miR-200 share PTEN as a targeting protein, so we added information and references about PTEN. Thank you for improving by editing as pointed out.

Lines 261-265: This only regards to senescence, i.e., dogs that are beyond the reproductive age (6-7 years depending on body weight), which is not the case for this research.

Answer: Although the study in humans does not apply equally in animals, it was cited to explain the decline in reproductive capacity with age in animals, especially dogs.

Lines 282-284: in which tissue? Testicular or epididymal miR-26a?

Answer: This section describes epididymal miR-26a.

Conclusion

Lines 348-349: please, be more specific on the molecular pathways involved with all these miRNAs.

Answer: I’m sorry for not being satisfied conclusions. It is not easy to draw conclusions to a specific molecular pathway with all these miRNAs in this manuscript. However, it was confirmed that miRNA26a and miR-200 share PTEN as a targeting protein, so we added specific molecule, PTEN.

Reviewer 3 Report

The authors of the manuscript entitled "Integrated miRNA in Canine Testis and Epididymis according 2 to Age, Cryptorchidism, and Testicular Tumor" investigated the expression of some miRNAs in dogs with cryptorchidism or testicular tumor, to identify possible valid markers for the evaluation of the physio-pathological state of the male dog reproductive system.

The subject is very topical given the roles that miRNAs are assuming in the veterinary field in a predictive, diagnostic and prognostic sense in some problems such as tumors.

The work presents some relevant concerns, that are following reported.

"Simple Summary" is too short and looks like a telegram. I suggest you read the instructions for the authors.

L. 50: you say that testicular tumors are benign. This is not true, some of them are malignant.

L. 74: you report that the diagnosis of tumor was made by histological examination. The question is: was a biopsy done preoperatively or by histological analysis?

In materials and methods, experimental design is not described in an intelligible way. For example, from which animals the testicle was taken and from whom the epididymis.

L. 96: "RNA later" should be one word.

L. 141: "creatine" should be "creatinine". To be corrected for the whole work, including table 1.

In the paragraph "clinical data" 3.1. the data reported are already in table 1, so it is useless to rewrite them.

Table 1: in the "Age, month" row, it is reported that the age of dogs with cryptorchidism, with Sertoli cell tumor and seminoma is 7.8; 10; and 12 months old, respectively? Is it right?

In the results, comparison between less than one year and more than 3 years old dogs has been done only for the comparison between testis and epididymis in normal dogs and in all other experiments the comparison is with less than one year old dogs including cryptorchidism and tumors. Could it better to compare with older normal dogs?

The level of up or downregulation of microRNAs has not been tested in older dogs. Especially cryptorchidism, the risk of tumor formation increases when the dog is old. In the paper the level of microRNAs of cryptorchid dogs has been done in less than one year old dogs, while it should be tested in older dogs.

L. 331 and 332 of discussion must be changed because they have mixed the result in dogs and human in an unclear way.

In the last paragraph of discussion, they talk about limitations for example age but they have compared ages only for testis and epididymis in normal animals and nothing else. The level of expression of microRNAs in cryptorchidism and tumors in older than 3 years is not reported in this paper. For both of these ages tis would be very important.

Author Response

The authors of the manuscript entitled "Integrated miRNA in Canine Testis and Epididymis according 2 to Age, Cryptorchidism, and Testicular Tumor" investigated the expression of some miRNAs in dogs with cryptorchidism or testicular tumor, to identify possible valid markers for the evaluation of the physio-pathological state of the male dog reproductive system.

The subject is very topical given the roles that miRNAs are assuming in the veterinary field in a predictive, diagnostic and prognostic sense in some problems such as tumors.

The work presents some relevant concerns, that are following reported.

Answer: First of all, thank you very much for your good points and advice. I modified it according to your detailed and important advice.

"Simple Summary" is too short and looks like a telegram. I suggest you read the instructions for the authors.

Answer: As you pointed out, I re-checked the instructions for the authors and I added content and refined the terms and sentences.

 → MicroRNA has been actively studied for diagnostic, prognostic, and therapeutic purposes of various diseases in animals. In this study, we analyzed miRNA changes according to various conditions in testis and epididymis of male dogs. Differentially expressed miRNAs in testis and epididymis were discovered depending on age and cryptorchidism in male dogs suggesting that they could be important potentials for understanding aging and disease in male dog reproductive system.

L.50: you say that testicular tumors are benign. This is not true, some of them are malignant.

Answer: There was an error in the content description. It's not benign, it's malignant. I modified it.

L.74: you report that the diagnosis of tumor was made by histological examination. The question is: was a biopsy done preoperatively or by histological analysis?

Answer: All of the histological tests of the tumors used in this study consisted of the entire tissue extracted after surgery, and the preoperative fine needle aspiration test was not performed due to accuracy problems that did not represent the entire tissue.

In materials and methods, experimental design is not described in an intelligible way. For example, from which animals the testicle was taken and from whom the epididymis.

Answer:  Of the 19 dogs used in this study, testicles and epidididymis were collected and analyzed separately except for two testicular tumor dogs. In two testicular tumor dogs, epidididymis were not analyzed separately for the analysis of the testis itself. I added this information to the M&M section that you pointed out.

 → All testes were obtained from all 19 male dogs, and epididymis cauda tissues (tail part of epididymis) were collected from 17 male dogs except for two testicular tumor dogs during castration at the request of owners with informed consent from the owner.

L.96: "RNA later" should be one word.

Answer: The spacing is incorrect. I modified it and added information about the reagent.

 → RNAlater (Thermofisher, Waltham,MA, USA)

L.141: "creatine" should be "creatinine". To be corrected for the whole work, including table 1.

Answer: I corrected all the terms that I made a mistake. Creatine → creatinine

In the paragraph "clinical data" 3.1. the data reported are already in table 1, so it is useless to rewrite them.

Answer: we move them into M&M.

Table 1: in the "Age, month" row, it is reported that the age of dogs with cryptorchidism, with Sertoli cell tumor and seminoma is 7.8; 10; and 12 months old, respectively? Is it right?

Answer: There was a mistake in unifying the format of the table. It is not 10 months or 12 months old, but 10 years old and 12 years old, so I revised this. The age of cryptorchidism dogs is 7.8 months as written.

 → 120 months / 144 months

In the results, comparison between less than one year and more than 3 years old dogs has been done only for the comparison between testis and epididymis in normal dogs and in all other experiments the comparison is with less than one year old dogs including cryptorchidism and tumors. Could it better to compare with older normal dogs?

Answer: In the case of cryptorchidism dogs, a diagnosis can be made on 6 months of old. Therefore, it was compared using immature dogs below 1 year old dogs as a control group. Their average old is 5.9±0.7 months old.

The level of up or downregulation of microRNAs has not been tested in older dogs. Especially cryptorchidism, the risk of tumor formation increases when the dog is old. In the paper the level of microRNAs of cryptorchid dogs has been done in less than one year old dogs, while it should be tested in older dogs.

Answer: In fact, in recent years in our country, it was not easy to meet the cryptorchidism of mature dogs (older than 1 year) because owners do not leave their retained testes neglected and perform preventive or therapeutic neutralization surgery when they are young. I know what you have pointed out is correct and future studies plan to recruit individuals over a long period of time to do additional research on adult dogs with latent testes. In this study, the number of testicular tumor patients was also insufficient, so we are planning and collecting additional research plans in the long term.

L.331 and 332 of discussion must be changed because they have mixed the result in dogs and human in an unclear way.

Answer: Do you mean in 352~353 about miR-497? we didn’t intend to mix the result in dogs and human. We only noted that the miRNAs seen infertile men were consistent with our study, but did mix the results. I’m sorry for confusing.

In the last paragraph of discussion, they talk about limitations for example age but they have compared ages only for testis and epididymis in normal animals and nothing else. The level of expression of microRNAs in cryptorchidism and tumors in older than 3 years is not reported in this paper. For both of these ages tis would be very important.

Answer: As mentioned above, this study acknowledges that there was a limit to the diversity of samples. So we've described it as limitations, and to make up for it, we're recruiting cryptorchidism at various ages, especially adult dogs, and a variety of tumor patients. I plan to supplement the current paper in the next paper.

Round 2

Reviewer 2 Report

The manuscript has improved considerably. Authors have made changes in the Introduction and Material and Methods sections to provide more information on the experimental design. Additionally, substantial modifications have been performed throughout.

Further comments are as follow:

If you want to keep the data of testicular tumors, you should clearly state that this is a preliminary study on miRNA expression in testicular tumoral tissue and therefore interpretation of data needs to be done with caution.

Abstract

Lines 31-32: please state clearly the type of relationship between miRNAs and testicular disorders you may infer with your obtained data.

Introduction:

Line 74: “testis and epididymis” instead of “reproductive tissues”

Line 75: the objective of using miRNA of therapeutic targeting has not been performed directly in this research. Therefore, this aim should be stated as future prospectives.

Discussion

Line 247: “the impact of age, cryptorchidism, and, preliminarily, testicular tumors on miRNA expression…”

Author Response

The manuscript has improved considerably. Authors have made changes in the Introduction and Material and Methods sections to provide more information on the experimental design. Additionally, substantial modifications have been performed throughout.

 Answer: First of all, I deeply appreciate your continuous good advice and help our paper become a good paper. As you advised me last time, I revised it as much as possible, and I will continue to revise it faithfully as you added. Thank you.

Further comments are as follow:

If you want to keep the data of testicular tumors, you should clearly state that this is a preliminary study on miRNA expression in testicular tumoral tissue and therefore interpretation of data needs to be done with caution.

Answer: As you advised, we added the following sentence to the discussion so that it can be clearly stated.

The analysis of miRNA expression in testicular tumor tissue conducted in this study is a preliminary study of a more extensive and population-expanding full-scale study. Therefore, the interpretation of the resulting data in this part should be approached carefully.

Abstract

Lines 31-32: please state clearly the type of relationship between miRNAs and testicular disorders you may infer with your obtained data.

Answer: In the end, this study revealed the cause-and-effect relationship between miRNA and aging and cryptorchidism, and we revised it to the following sentence.

This study demonstrated that aging and cryptorchidism have a causal relationship with miRNA expression.

Introduction:

Line 74: “testis and epididymis” instead of “reproductive tissues”

Answer: As you pointed out, I wrote it by limiting “reproductive tissues” to “testis and epididymis” according to the scope of the paper.

Line 75: the objective of using miRNA of therapeutic targeting has not been performed directly in this research. Therefore, this aim should be stated as future prospectives.

Answer: As you said, I revised the content to mean that it suggests future therapeutic possibilities.

The purpose of this study is to find the relationship between aging and disease status and expression of miRNA in male dog testis and epididymis and to suggest that it can be used as biomarkers and therapeutic targets in the future for infertility and disease prevention in male dogs.

Discussion

Line 247: “the impact of age, cryptorchidism, and, preliminarily, testicular tumors on miRNA expression…”

Answer: Thank you for correcting the tumor part in that sentence so that it doesn't matter. I revised it the same as you told me.

Thank you very much.

Reviewer 3 Report

The authors have responded extensively to many comments, but I would like to return to some of them for further clarification.

L. 88: I agree when you say that fine needle aspiration is not an accurate investigation like the one performed on whole tissue, but I would recommend adding a sentence like this: tumor diagnosed by histological examination on surgically removed tissue.

Regarding the paragraph "Clinical data", I apologize if I was not clear: I meant that if the data are reported in the table, it is useless and redundant if the same data are also reported in the text. I had advised to rewrite this part, eliminating the non-significant data, thus reporting only the most significant ones. And to report them in the "Results".

Author Response

The authors have responded extensively to many comments, but I would like to return to some of them for further clarification.

Answer: First of all, I deeply appreciate your continuous good advice and help our paper become a good paper. As you advised me last time, I revised it as much as possible, and I will faithfully revise the additional advice you gave me this time. Thank you.

L. 88: I agree when you say that fine needle aspiration is not an accurate investigation like the one performed on whole tissue, but I would recommend adding a sentence like this: tumor diagnosed by histological examination on surgically removed tissue.

Answer: Thank you for giving me a clear sentence about the diagnosis part. As you advised, I connected the sentences smoothly.

Testicular tumor diagnosis was based on histological examination (IDEXX, Republic of Korea) on surgically removed tissue.

Regarding the paragraph "Clinical data", I apologize if I was not clear: I meant that if the data are reported in the table, it is useless and redundant if the same data are also reported in the text. I had advised to rewrite this part, eliminating the non-significant data, thus reporting only the most significant ones. And to report them in the "Results".

Answer: In fact, another reviewer advised that all the clinical data part should be transferred to M&M part. I deleted the unnecessary parts overall because it overlaps with the contents of the table you mentioned, and I made and modified a sentence that compresses the contents of the objects used in this study.

We obtained eight healthy male puppies under the age of one-year old and four healthy male dogs over the age of three, five cryptorchidism dogs under the age of one-year, one dog with sertoli cell tumor and one dog with seminoma over the age of ten. The results of each item of their average serum chemistry and the reference ranges of each serum chemistry are described in the Table1.

Thank you very much.